# Intracellular hydrogelation preserves fluid and functional cell membrane interfaces for biological interactions

Jung-Chen Lin[1], Chen-Ying Chien[1], Chi-Long Lin[1], Bing-Yu Yao[1], Yuan-I Chen[1], Yu-Han Liu[1], Zih-Syun Fang[1,2], Jui-Yi Chen[1], Wei-ya Chen[1], No-No Lee[1,2], Hui-Wen Chen[2] & Che-Ming J. Hu [1]

Cell membranes are an intricate yet fragile interface that requires substrate support for stabilization. Upon cell death, disassembly of the cytoskeletal network deprives plasma membranes of mechanical support and leads to membrane rupture and disintegration. By assembling a network of synthetic hydrogel polymers inside the intracellular compartment using photo-activated crosslinking chemistry, we show that the fluid cell membrane can be preserved, resulting in intracellularly gelated cells with robust stability. Upon assessing several types of adherent and suspension cells over a range of hydrogel crosslinking densities, we validate retention of surface properties, membrane lipid fluidity, lipid order, and protein mobility on the gelated cells. Preservation of cell surface functions is further demonstrated with gelated antigen presenting cells, which engage with antigen-specific T lymphocytes and effectively promote cell expansion ex vivo and in vivo. The intracellular hydrogelation technique presents a versatile cell fixation approach adaptable for biomembrane studies and biomedical device construction.

---

[1] Institute of Biomedical Sciences, Academia Sinica, Taipei 11574, Taiwan. [2] Department of Veterinary Medicine, National Taiwan University, Taipei 10617, Taiwan. These authors contributed equally: Jung-Chen Lin, Chen-Ying Chien, Chi-Long Lin. Correspondence and requests for materials should be addressed to C.-M.H. (email: chu@ibms.sinica.edu.tw)

The cell membrane is a fluid substrate that harbors a milieu of phospholipids, proteins, and glycans, which dynamically choreograph numerous biological interactions. The long-standing fascination with the various biological functions of cell membranes has inspired model systems and cell-mimetic devices for biological studies[1–3], tissue engineering[4,5], drug delivery[6–8], and immunoengineering[9–12]. Toward replicating the cell membrane interface, synthetic bilayer lipid membranes and bioconjugation strategies are commonly adopted in bottom-up engineering of cell membrane mimics[13]. Alternatively, top-down approaches based on extraction and reconstitution of plasma membranes of living cells are frequently applied to capture the intricate cell-surface chemistries for biomimetic functionalization[6–8]. As antigen presentation, membrane fluidity, and membrane sidedness are critical factors behind biomembrane functions and can be influenced by membrane translocation processes, methods for harnessing this membranous component continue to emerge with the aim to better study and utilize this complex and delicate biological interface[14–16].

To stabilize the fluid and functional plasma membranes and decouple it from the dynamic state of living cells, we envision that a synthetic polymeric network can be constructed in the cytoplasm to replace the cytoskeletal support for stabilizing cellular structures. Unlike endogenous cytoskeletons that are susceptible to reorganization and disintegration upon perturbation and cell death[17], a synthetic substrate scaffold can stably support the cell membrane interface for subsequent applications. As the mechanical property of cytoskeletons has drawn comparisons to hydrogels[17,18], a cellular fixation approach mediated by intracellular assembly of hydrogel monomers is herein developed. We demonstrate that the intracellular hydrogelation technique effectively preserves cellular morphology, lipid order, membrane protein mobility, and biological functions of the plasma membrane, giving rise to cell-like constructs with extraordinary stability. In addition, a highly functional artificial antigen presenting cell (APC) is prepared with the gelated system to highlight the platform's utility for biomedical applications.

## Results

### Intracellular hydrogelation by photoactivated cross-linking.
Three criteria were considered to establish the intracellular hydrogelation technique: (i) Hydrophilic cross-linking monomers with a low-molecular weight were used to facilitate cytoplasmic permeation and minimize membrane partitioning. (ii) Cross-linking chemistry with low-protein reactivity was adopted to facilitate nondisruptive cellular fixation. (iii) Extracellular cross-linking was minimized to prevent cell-surface masking. Based on these considerations, a photoactivated hydrogel system consisting of poly(ethylene glycol) diacrylate monomer (PEG-DA; $M_n$ 700) and 2-hydroxyl-4′-(2-hydroxyethoxy)-2-methylpropiophenone photoinitiator (I2959) was employed. The materials are broadly used in biomedical applications and have little reactivity with biological components[19,20]. These hydrogel components were introduced into cells through membrane poration with a single freeze–thaw cycle. Following a centrifugal wash to remove extracellular monomers and photoinitiators, the cells were irradiated with ultraviolet (UV) light for intracellular hydrogelation (Fig. 1a and Supplementary Fig. 1). To assess the feasibility of intracellular gelation for cellular fixation, HeLa cells were first processed with different PEG-DA cross-linker densities ranging from 4 to 40 wt%. The freeze–thaw treatment allowed PEG-DA monomers to penetrate into the intracellular domain efficiently, and the collected cells had PEG-DA contents equivalent to the input PEG-DA concentrations (Fig. 1b). Following UV irradiation to the PEG-DA infused cells, no alteration to the cellular

morphology was observed (Supplementary Fig. 2). An evaluation by atomic force microscopy, however, showed that the gelated cells (GCs) exhibited increasing Young's moduli that correlated with the PEG-DA concentrations (Fig. 1c). Assessment of GC stability by microscopy showed no observable structural alternation over a 30-day observation period, whereas control cells and non-crosslinked cells exhibited noticeable disintegration within 3 days (Fig. 1d and Supplementary Fig. 3). To further confirm the assembly of hydrogel networks in the intracellular domain, fluorescein-diacrylate was added to the cross-linker mixture to covalently imbue the hydrogel network with green fluorescence (Supplementary Fig. 1). Following membrane staining with a lipophilic DiD fluorophore, GCs showed distinctive membranous and hydrogel components (Fig. 1e and Supplementary Fig. 4), displaying a structure reminiscent of substrate-supported lipid membranes[13]. Solubilization treatment with sodium dodecyl sulfate was applied to examine the integrity of the gelated cytoplasm, and the fluorescent hydrogel matrices in GCs remained intact following membrane dissolution (Supplementary Fig. 4). In a dye-exclusion study, 4 wt% GCs effectively excluded a water-soluble fluorescein isothiocyanate (FITC) dye from entering the cytoplasm (Fig. 1f and Supplementary Fig. 4), thereby confirming the plasma membrane integrity on GCs. We also demonstrated that GCs could be stored by freezing and by lyophilization (Supplementary Fig. 5A). In addition, the intracellular gelation process was applied to adherent HeLa cells, effectively preserving the cells' adherent property and elongated structures (Supplementary Fig. 5B).

### Intracellular gelation preserves cellular features.
Examination of the cell membrane interface and the cytoplasmic hydrogel matrix on GCs was performed by transmission electron microscopy (TEM). In comparison to control cells, GCs possessed a perforated, hydrogel-filled interior. Treatment by detergent stripped GCs of their membranous exterior, leaving behind nondissolvable hydrogel matrices (Fig. 2a). To better visualize the membrane interface on GCs, intracellular hydrogelation was applied to avian red blood cells (aRBCs), which are nucleated cells devoid of organelles. As hemoglobins were removed during the gelation process, gelated aRBCs (G-aRBCs) exhibited a clear membrane boundary encircling a perforated nucleus (Fig. 2b and Supplementary Fig. 6). Notably, the addition of a hemagglutinating influenza virus to the G-aRBCs induced direct agglutination (Supplementary Fig. 6), and TEM cryosections showed similar binding patterns between nongelated aRBCs and G-aRBCs (Fig. 2c and Supplementary Fig. 6). These results highlight the intracellular hydrogelation technique enables facile preparation of stable, cell-like constructs without masking cellular surfaces. Using adherent HeLa cells, we further assessed the influence of intracellular hydrogelation on periplasma components and cellular cytoskeletons. In contrast to live cells that lost membrane ruffles upon incubation in phosphate-buffered saline (PBS) for 4 h, both 4 wt% and 20 wt% GCs in PBS retained the ruffled membrane features (Fig. 2d). Confocal microscopy of actin-GFP-transfected HeLa cells also showed that filamentous actin structures were observable in the GCs (Fig. 2e and Supplementary Fig. 7A). Interestingly, in 20 and 40 wt% GCs, actin filaments could be observed 24 h after the gelation process, which suggests that the denser hydrogel matrices could entrap these intracellular components and retard their depolymerization and dissipation (Supplementary Fig. 7A). We also observed that GCs could retain their ruffled exterior over a long period of time (Supplementary Fig. 7B, C), further illustrating that the synthetic hydrogel networks can substitute cytoskeleton in supporting these nanoscale membrane features.

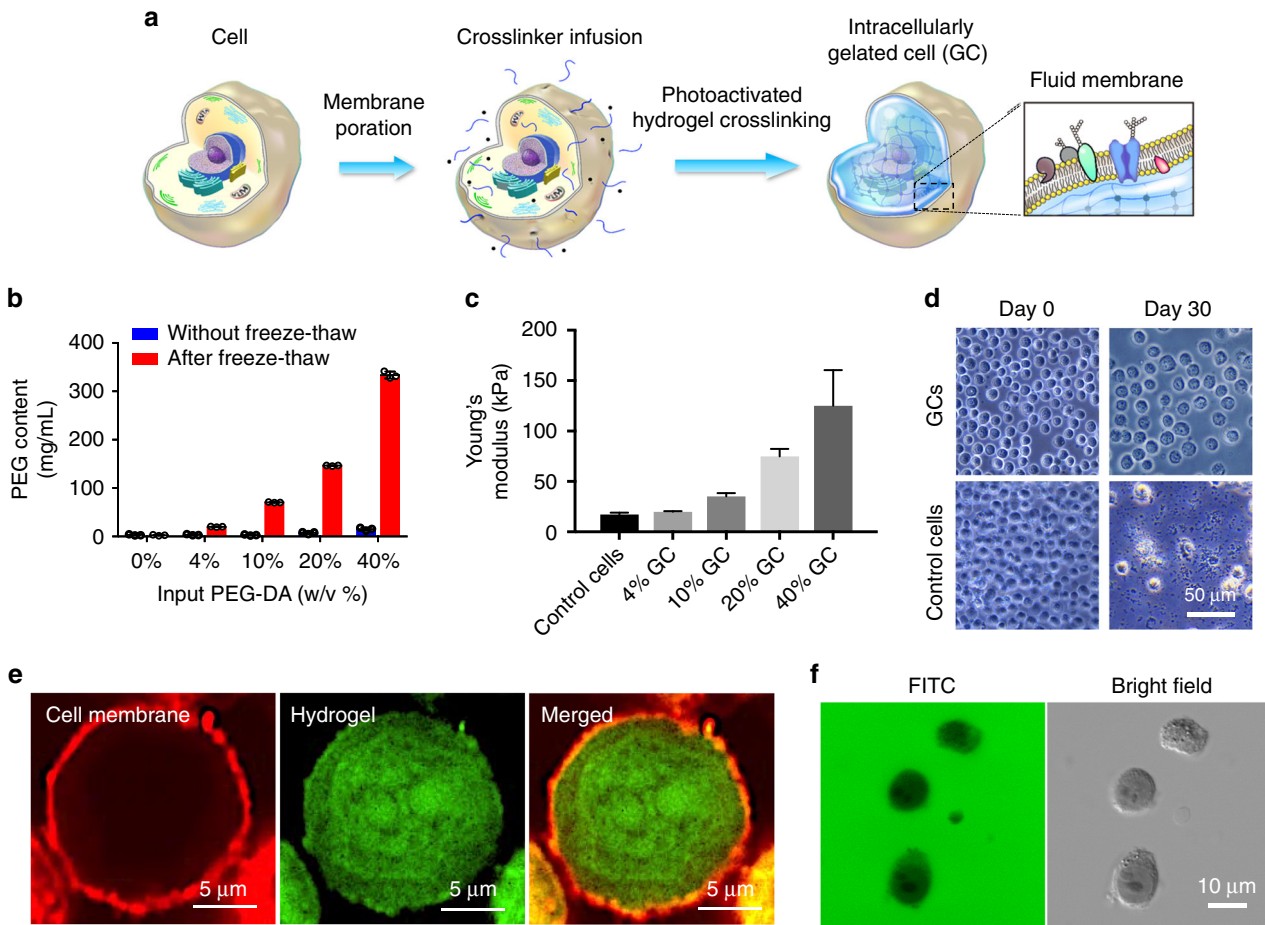

**Fig. 1** Preparation and characterization of intracellularly gelated cells (GCs). **a** Hydrogel monomers and photoinitiators are infused into the intracellular domain of cells following transient membrane poration. UV-activated hydrogel cross-linking is then performed to stabilize the cell membrane interface. **b** Intracellular concentrations of PEG in cells before and after freeze–thaw treatments in gelation buffers containing different PEG-DA content. Error bars represent mean ± standard deviation, $n = 3$. **c** The Young's moduli of the GCs prepared with different concentrations of hydrogel monomers were measured by atomic force microscopy. Error bars represent mean ± standard deviation, $n = 64$. **d** Bright-field microscopy of 4 wt% GCs and control cells suspended in PBS for 0 and 30 days. Scale bar = 50 μm. **e** Structure of 20 wt% GCs was visualized with fluorescein-diacrylate (green) for hydrogel labeling and DiD dye (red) for membrane staining. Scale bars = 5 μm. **f** 4 wt% GCs suspended in fluorescein solution showed that the GCs were impermeable to the dye. Scale bar = 10 μm

**Intracellular gelation preserves lipid order and fluidity**. We next examined the influence of intracellular hydrogelation and hydrogel densities on plasma membrane fluidity and membrane lipid order on GCs (Fig. 3a). Assessment of membrane fluidity by fluorescence recovery after photobleaching (FRAP) using a lipophilic DiD dye showed that fluorescence recovery halftimes were similar among live cells and GCs of different cross-linking densities (Fig. 3b, c and Supplementary Fig. 8), indicating that the intracellular hydrogel matrices did not influence membrane lipid fluidity regardless of the hydrogel content. Given that membrane order is a critical biophysical parameter that influences the dynamics of membrane proteins, we adopted a Laurdan dye staining approach to quantify membrane order on GCs[21]. Through multiphoton microscopy followed by image processing to analyze the polarity of the plasma membrane, we were able to distinguish the ordered plasma membrane in live HeLa cells and derive the corresponding generalized polarization (GP) values by tracing the pixel intensities at the cellular periphery (Fig. 3d). Upon applying the technique to GCs and control cells subject to freeze–thaw and UV treatments in the absence of hydrogel, preservation of plasma membrane order by intracellular gelation was confirmed. Whereas nongelated control cells showed

noticeable alteration in membrane order and GP values, membrane order in GCs of different hydrogel densities was similar to that of live cells (Fig. 3e, f). These results demonstrate that the gelation process has little influence on the phospholipid bilayer, effectively retaining the membrane fluidity and membrane order in the stabilized GCs.

**Membrane proteins retain lateral mobility on GCs**. To evaluate the mobility of membrane proteins on GCs, we first chose CD80 as the protein of interest given that CD80's putative T-cell stimulating functionality is highly dependent on its lateral mobility[3]. CD80-GFP mobility on GCs was first assessed with total internal reflection fluorescence (TIRF) microscopy, which revealed rapid, random movements of fluorescent punctates. In contrast, fluorescent signals in glutaraldehyde-fixed cells appeared static (Fig. 4a, Supplementary Fig. 9, and Supplementary Movies 1–5). As live cells showed prominent cytoskeleton-directed protein movements owing to actin-mediated CD80 localization[22,23], cells treated with a single freeze–thaw cycle were freshly prepared as a control (Supplementary Movies 6 and 7). Calculation of protein diffusivity showed a mean value of 0.118 μm$^2$ s$^{-1}$ for the control cells, which is in accordance with prior studies on the passive

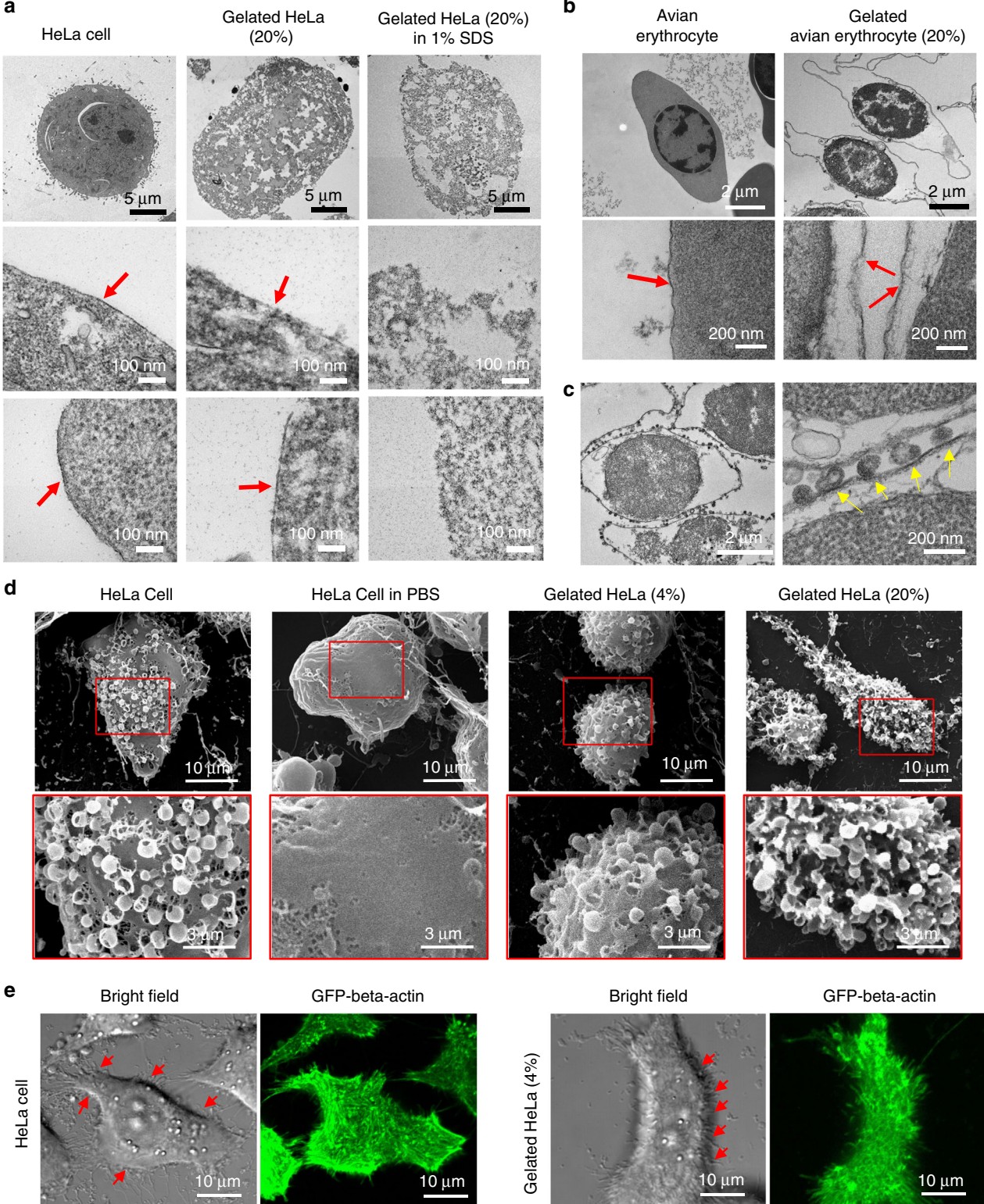

**Fig. 2** Structural examination of GCs. **a** Transmission electron microscopy (TEM) images show the cross-sectional structure of normal HeLa cells (left), 20 wt% gelated HeLa cells (middle), and the hydrogel matrix of 20 wt% gelated HeLa cells following solubilization by 1% SDS (right). (scale bars = 5 µm or 100 nm). Red arrows indicate cell membranes. **b** TEM cryosection images show the structure of avian erythrocytes and gelated avian erythrocytes (20 wt% PEG-DA) and **c** binding of hemagglutinating influenza viruses on the surfaces of gelated avian erythrocytes (scale bars = 2 µm or 200 nm). Red arrows indicate cell membranes. Yellow arrows indicate influenza viruses. **d** Cryogenic scanning electron microscopy (Cryo-SEM) images show the surface features of live HeLa cells, HeLa cells in PBS for 4 h, 4 wt% gelated HeLa cells in PBS, and 20 wt% gelated HeLa cells in PBS. Membrane ruffles were observed on the gelated cells. Scale bars = 10 µm (top row) and 3 µm (bottom row). **e** HeLa cells transfected with plasmids carrying actin-GFP and the corresponding 4 wt% GCs were imaged under bright-field and fluorescence microscopy. Preservation of actin filament was observed in the GCs. Red arrows indicate membrane ruffles, scale bars = 10 µm

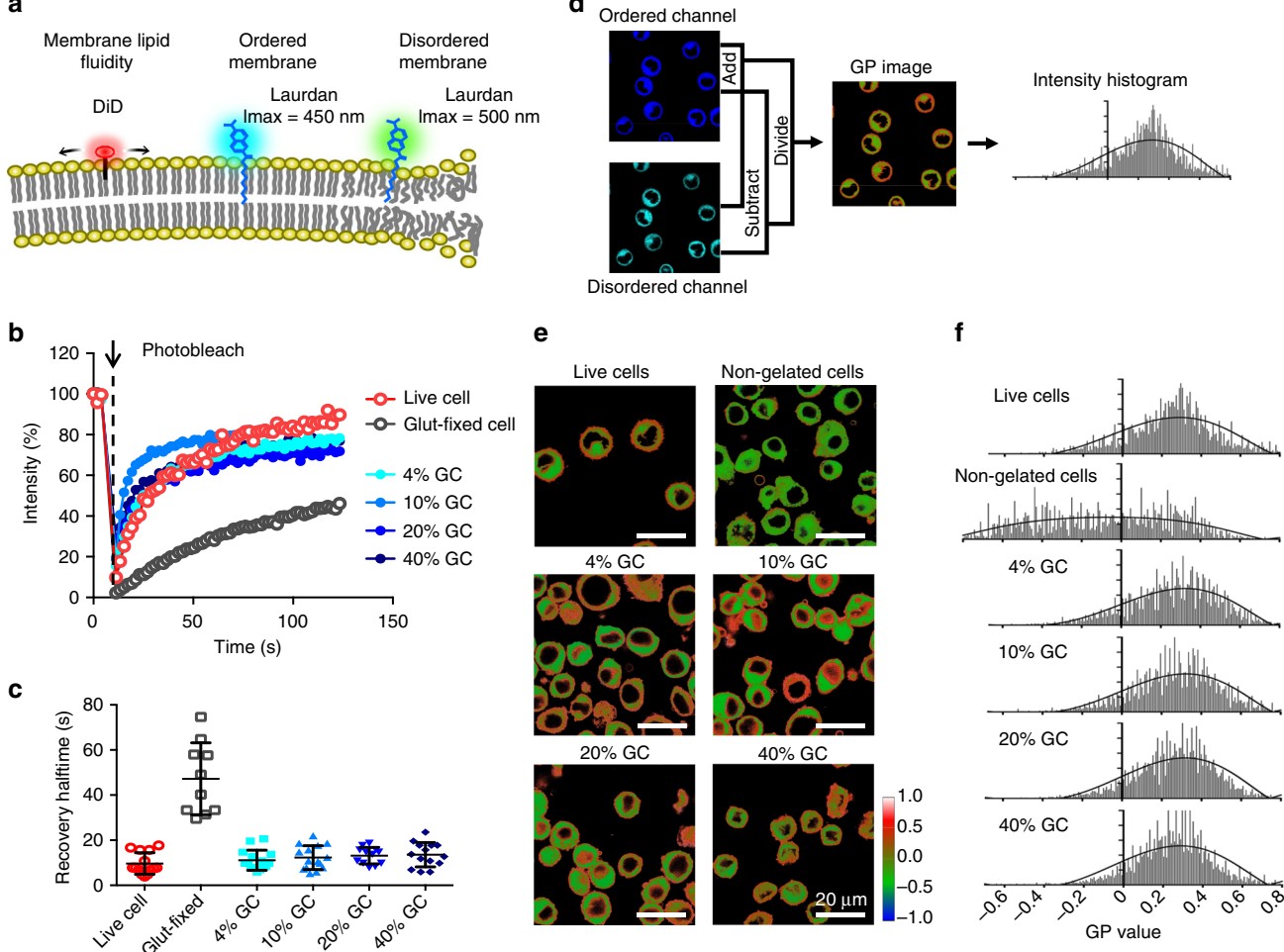

**Fig. 3** Examination of lipid fluidity and membrane order on the plasma membrane of GCs. **a** A schematic illustration of the fluorescent probes used for assessing lipid fluidity (DiD) and membrane order (Laurdan). **b** Representative fluorescence recovery curves after photobleaching of DiD dye on HeLa cells, glutaraldehyde-fixed cells, and GCs of different hydrogel densities. **c** Halftimes of DiD dye fluorescence recovery following photobleaching for live HeLa cells, glutaraldehyde-fixed HeLa cells, and gelated HeLa cells with different hydrogel densities. No significant difference was observed between each of the GCs and live cells. Error bars represent mean ± standard deviation, ($n = 10$–$15$). **d** A flowchart illustrating the derivation of generalized polarization (GP) values from confocal images in the ordered and disordered channels. **e** Representative pseudo-colored GP-intensity-merged images of live HeLa cells, nongelated control cells, and GCs of different hydrogel densities. Ordered membrane domains are shown in orange. Scale bars = 20 μm. Color scale corresponds to GP value. **f** Histograms of GP values comparing the membrane order of the plasma membrane on live cells, nongelated control cells and GCs

diffusion of transmembrane proteins[24]. For the 4, 10, 20, and 40 wt% GCs, the mean diffusivities were 0.118, 0.0966, 0.0967, and 0.0711 μm$^2$ s$^{-1}$, respectively (Fig. 4b). No statistical significance, however, was observed among the control cell and the GCs.

Further assessment of protein mobility was performed with FRAP on adherent GCs. After photobleaching, rapid fluorescence recovery further validated the fluidity of CD80-GFP on GCs (Fig. 4c; Supplementary Fig. 10). Curiously, close examination of the GCs by Z-stacked fluorescence microscopy revealed distinctive fluorescent filaments consistent with the patterns of actin cytoskeleton (Supplementary Fig. 10D). Given CD80's tendency to complex with actin[22,23], the observed filamentous patterns can be attributed to the actin/CD80 complexes. To minimize interference by cytoskeleton-directed protein movements and intracellular protein trafficking in comparing protein mobility between GCs and live cells, a first-order kinetics equation was applied to the early time points of the fluorescence recovery curves to derive fluorescence recovery kinetics. Based

on the kinetics analysis, it was confirmed that GCs of varying densities possessed similar CD80-GFP recovery rates to those of live cells (Fig. 4e; Supplementary Fig. 10B, C). In contrast, glutaraldehyde-fixed cells showed significantly reduced recovery kinetics, highlighting intracellular hydrogelation as a unique approach for preserving mobile membrane proteins.

To evaluate how the hydrogel density may influence the mobility of different membrane proteins, HeLa cells transfected with glycosylphosphatidylinositol (GPI)-anchored enhanced green fluorescent protein (EGFP-GPI), transferrin receptor (TfR), GFP-tagged tyrosine protein kinase Lyn (Lyn-GFP), and GFP-tagged epidermal growth factor receptor (EGFR-GFP) were separately prepared for FRAP analysis (Fig. 4d). These proteins cover a broad range of differently sized cytoplasmic domains, with EGFP-GPI and TfR possessing no and small cytoplasmic segments and Lyn-GFP and EGFR-GFP having large cytoplasmic regions. Examination with FRAP showed that all the assessed proteins had significantly higher mobile fractions in GCs as compared to glutaraldehyde-fixed cells (Fig. 4e and

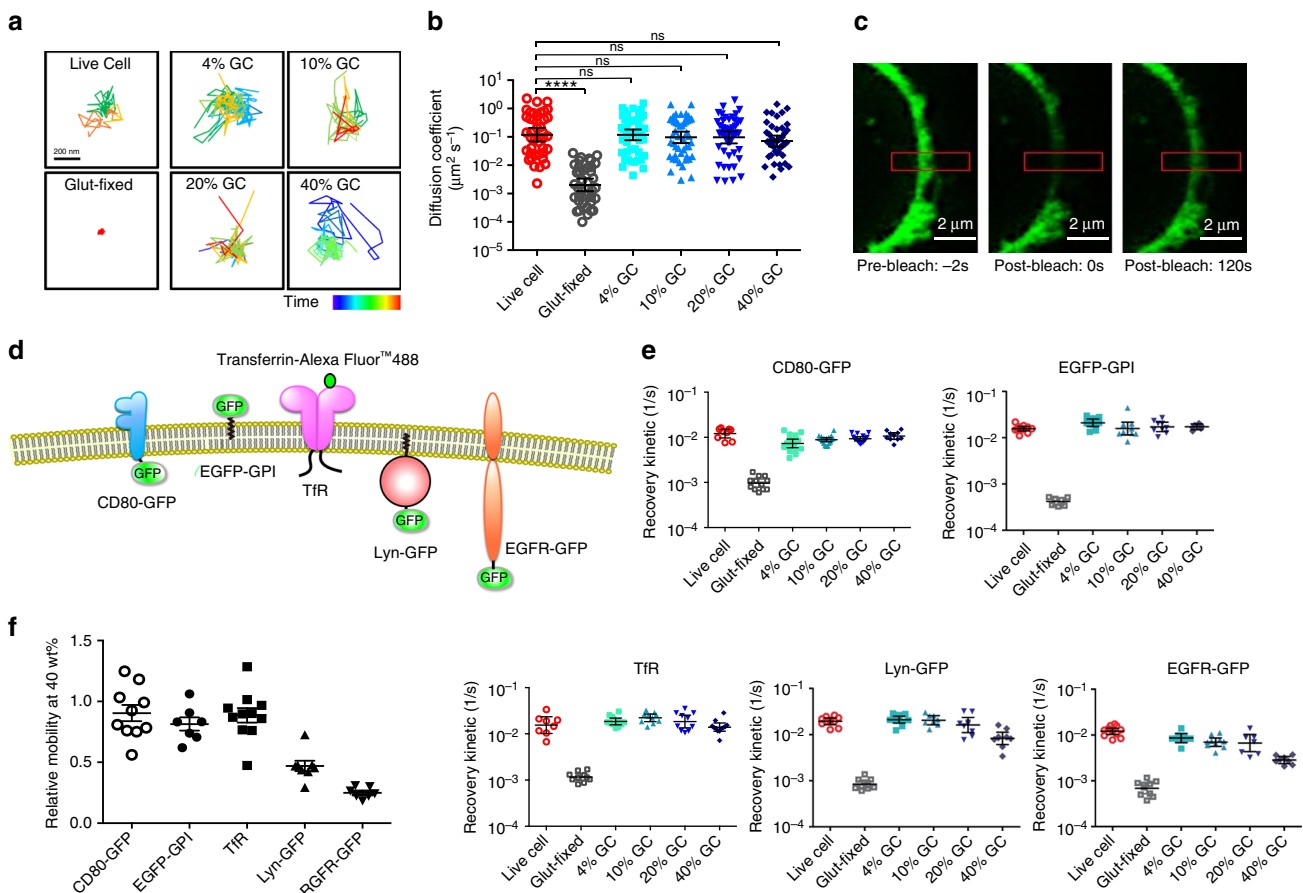

**Fig. 4** Examination of membrane protein lateral mobility on GCs. **a** Representative trajectories from CD80-GFP fluorescence tracking on different GCs and control cells examined by TIRF microscopy. **b** Diffusion coefficients of CD80-GFP were calculated from the TIRF fluorescence tracking data. Error bars represent geometric mean with 95% confidence interval. **c** Representative images showing recovery of CD80-GFP fluorescence in 4 wt% GC following photobleaching. Red rectangles indicate the photobleached area of interest. Scale bars = 2 μm. **d** A schematic illustration of membrane proteins with different sizes of intracellular domains, including CD80-GFP, EGFP-GPI, TfR, Lyn-GFP, and EGFR-GFP, which were assessed for their lateral mobility on GCs. **e** Recovery kinetics of CD80-GFP, EGFP-GPI, TfR, Lyn-GFP and EGFR-GFP on GCs and control cells assessed by FRAP. For GCs with 4 to 20 wt% hydrogel densities, all examined membrane proteins had similar lateral mobility as on live cells. Error bars represent geometric mean with 95% confidence interval. **f** Membrane protein mobility on 40 wt% GCs relative to their corresponding mean mobility on live cells. Error bars represent geometric mean with 95% confidence interval ($n = 7$–12). Statistical analysis was performed using a two-tail Student $t$ test, ****$p < 0.001$

Supplementary Figs. 11–14). Notably, Lyn-GFP and EGFR-GFP exhibited reduced relative mobility and mobile fractions at 40 wt% hydrogel cross-linking (Fig. 4f and Supplementary Figs. 13 and 14). It can be reasoned that the dense hydrogel core at 40 wt% cross-linking imposes higher drag to transmembrane proteins and may also entrap proteins with large cytoplasmic domains, resulting in the reduced lateral mobility of Lyn-GFP and EGFR-GFP. For CD80-GFP, EGFP-GPI, and TfR, however, 40% hydrogel core had little influence on their relative mobility. In addition, at gelation densities between 4 and 20 wt%, recovery kinetics for all the examined membrane proteins were mostly similar to those of live cells, indicating that sufficiently porous hydrogel cores can be prepared for cellular stabilization with minimal impact on membrane protein mobility. Upon long-term storage of GCs at 4 °C, the presence of mobile membrane proteins remained readily detectable (Supplementary Fig. 15), further demonstrating the construct's robust stability.

**Gelated dendritic cells for antigen presentation**. To highlight the potential utility of GCs, we prepared gelated dendritic cells (G-DCs) and assessed its antigen presenting capability. Effective T-lymphocyte expansion by DCs hinges on the presence of

multiple, mobile, membrane-bound lymphocyte activation signals[25], and replicating these biological features remains a primary engineering objective in the development of artificial APC systems. In our system, we hypothesized that G-DCs could trigger T-cell expansion through MHC class I-TCR and CD80-CD28 interactions (Fig. 5a). To prepare G-DCs, DCs were first activated by subjecting JAWSII murine DCs to SIINFEKL peptide pulsing and lipopolysaccharide stimulation. G-DCs were then prepared using 4 wt% PEG-DA with both activated and nonactivated DCs (Supplementary Fig. 16A, B). DCs fixed with glutaraldehyde were prepared as a control. Flow cytometric analysis of DC surface markers showed the expression of H-2K[b]/SIINFEKL and CD80 were largely similar before and after the gelation process (Fig. 5b–e). Among the activated DC samples, expression of H-2K[b]/SIINFEKL complexes in live and gelated DCs were 98.2% and 89.5%, respectively ($p > 0.05$). Likewise, the live and gelated DCs showed comparable CD80 levels at 79.3% and 63.4% respectively ($p > 0.05$). The nonactivated live and gelated DCs also shared similar basal levels of H-2K[b]/SIINFEKL and CD80 expression, demonstrating that the cell-surface protein signature was effectively preserved following the gelation process. The interaction between G-DCs and T-lymphocytes was further

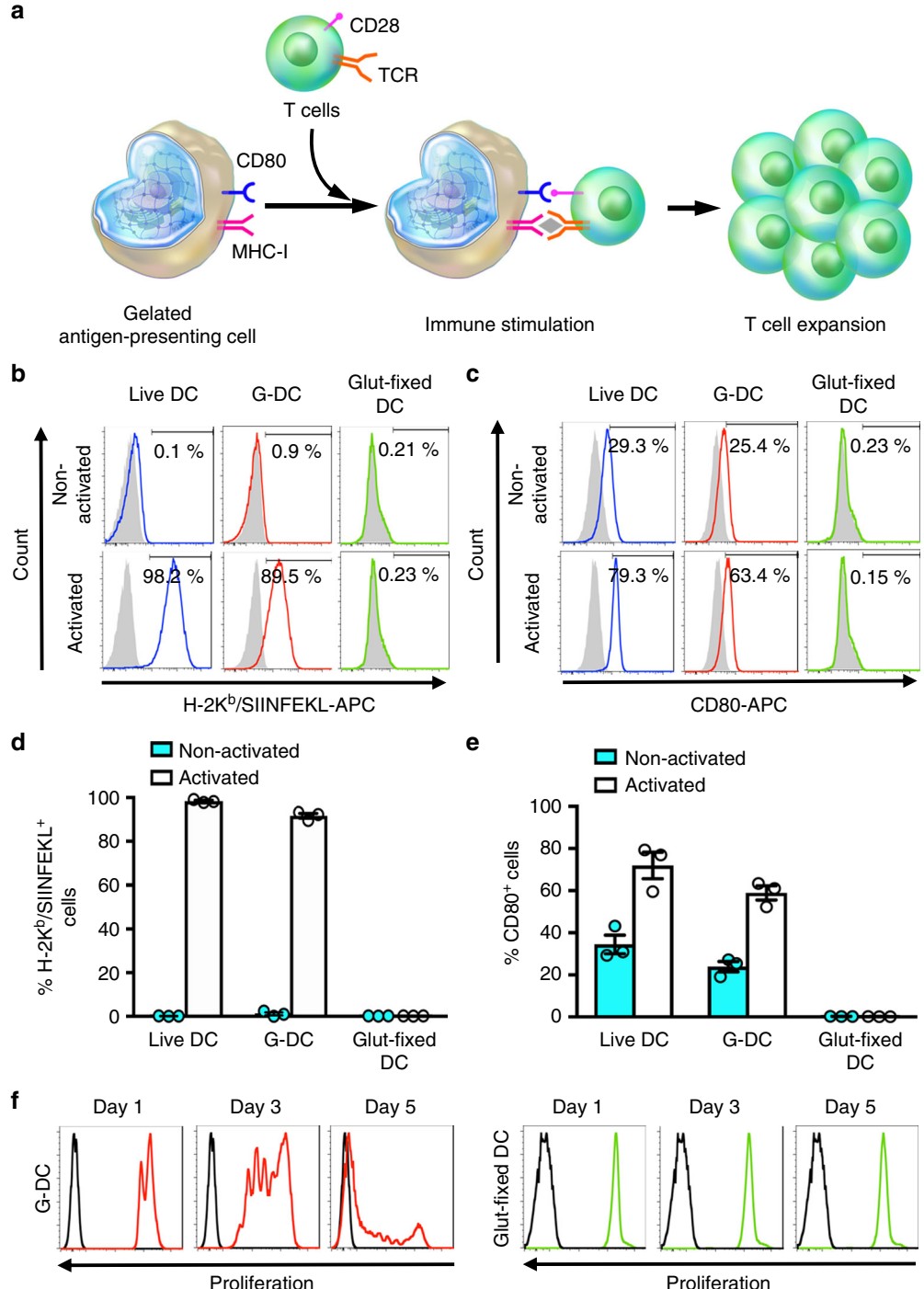

**Fig. 5** Gelated dendritic cells (G-DCs) as artificial antigen presenting cells. **a** A schematic illustration showing the interaction between T cells and G-DCs. JAWSII cells were treated with LPS and pulsed with SIINFEKL peptides for activation. Both nonactivated and activated cells were gelated for surface marker comparison. No significant difference was observed between the expression of **b**, **d** H-2K$^b$/SIINFEKL complexes and **c**, **e** CD80 on the surfaces of live DCs and G-DCs. Unstained T-cell groups (gray) were used as a negative control. Error bars represent mean ± SEM, $n = 3$. **f** Co-culture of activated G-DCs and glutaraldehyde-fixed DCs with CFSE-stained OT-I-specific CD8+ T cells showed time-dependent T-cell expansion by the G-DCs but not by the glutaraldehyde-fixed cells. Each culture condition contained a fixed number of DCs at $8 \times 10^4$ per well with a T-cell/G-DC ratio of 3:1. Unstained control T cells are plotted in black as a reference

examined by incubating CD8+ T cells derived from OT-I transgenic mice with activated G-DCs. The G-DCs effectively expanded the target T lymphocytes in a time- and cell ratio-dependent fashion (Fig. 5f and Supplementary Fig. 16C, D), validating preservation of functional cell membrane interface by intracellular hydrogelation.

**Gelated APCs dynamically interact with T cells**. T-cell interaction with activated G-DCs was visually examined through confocal microscopy. Upon incubation, multiple T cells were observed to interact with G-DCs actively, forming cell–cell conjugates (Fig. 6a and Supplementary Movie 8). Time-lapse imaging revealed continuous morphological changes and movements

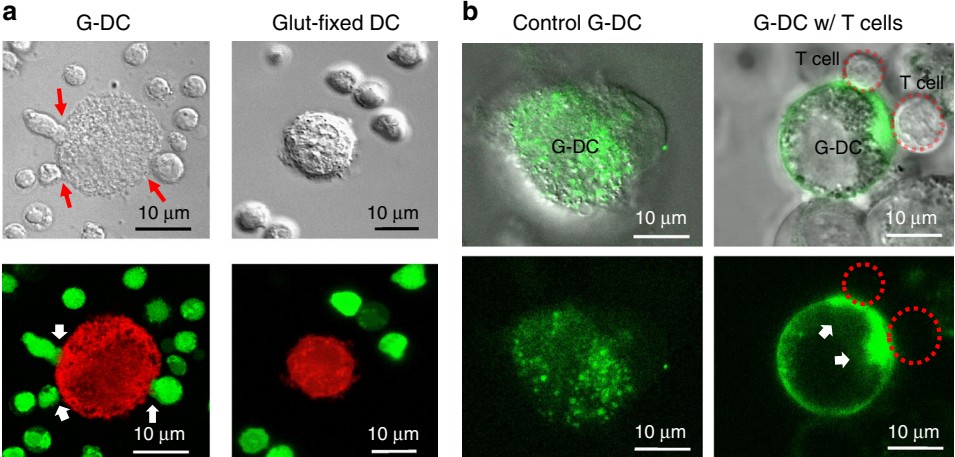

**Fig. 6** Examination of G-DC/T-cell interaction and CD80 clustering. **a** Incubation of CSFE-labeled CD8+ T cells (green) derived from OT-I transgenic mice and SIINFEKL-pulsed G-DCs (red) showed dynamic interaction between the T cells and G-DCs under bright-field and confocal fluorescence microscopy. Red and white arrows indicate cell-cell conjugates. Scale bars = 10 μm. **b** Incubation of CD80-GFP-expressing, SIINFEKL-pulsed G-DCs (green) and OT-I CD8+ T cells (red dashed-line circles) showed formation of CD80 clusters at the G-DC/T-cell junctions. White arrows indicate sites of CD80 clusters. Scale bars = 10 μm

among the T cells in apposition to the G-DCs, and each engagement event lasted approximately 2–3 h (Supplementary Movie 8). The interaction dynamic between the G-DCs and the T cells was similar to the many reports on live APCs and T cells[26,27], and this long-lasting yet nonstatic engagement has been deemed an essential process for immunological synapse formation and T-cell activation. Between glutaraldehyde-fixed DCs and T cells, no interaction was observed. We also examined CD80 distribution on G-DCs upon T-cell engagement by incubating CD80-GFP-transfected G-DCs with antigen-specific T cells. Following 3 h of incubation, fluorescent CD80 clusters were observed at G-DC/T-cell junctions (Fig. 6b and Supplementary Fig. 16E). Presence of these CD80 clusters attests to the fluid membrane interface on GCs and indicates that the T cells were able to sample and recruit these co-stimulatory signals on the G-DC surfaces. With CD80 clusters being a hallmark of immunological synapse[3], their presence helps justify the prominent T-cell expansion triggered by the G-DCs.

**Gelated APCs for ex vivo and in vivo T-cell expansion.** We then compared the antigen presentation capability of G-DCs to that of live DCs. An ex vivo T-cell proliferation assay using CFSE-labeled CD8+ OT-I T cells was first performed with G-DCs, live DCs, and glutaraldehyde-fixed DCs derived from the same cell source. Compared to live DCs that expanded the T cells by 78.7%, the G-DCs induced 56.2% of T-cell proliferation while lacking active T-cell stimulating processes such as cytokine secretion and microvilli formation[28]. In contrast, negligible T-cell expansion was observed for the glutaraldehyde-fixed DCs and nonactivated DC samples (Fig. 7a, b). The ability of G-DCs to stimulate T cells was further assessed in vivo with mice adoptively transferred with CFSE-labeled CD8+ OT-I T cells. Twenty-four hour following the adoptive transfer, mice were administered with 10^6 G-DCs, live DCs, or glutaraldehyde-fixed DCs. Notably, glutaraldehyde-fixed DCs induced asphyxiation and mortality shortly after injection, whereas G-DCs and live DCs were well tolerated. Flow cytometric analysis of the splenocytes 3 days following the DC injections revealed that both the activated G-DCs and live DCs induced substantial expansion of CD8+ T cells (Fig. 7c, d). This T-cell stimulation was antigen-specific as nonactivated DCs and G-DCs failed to induce T-cell division. To further highlight the role of hydrogel support in preserving the

cell membrane functionality, activated G-DCs and nongelated DCs were kept in PBS for 21 days at 4 °C. On day 21, the G-DCs, which retained their spherical morphology (Supplementary Fig. 16B), ably triggered antigen-specific T-cell expansion in vivo (Fig. 7e, f). In contrast, nongelated DCs showed prominent disintegration and had significantly reduced capacity in expanding T cells. In addition, G-DCs mechanically disrupted by ultrasonication resulted in reduced T-cell expansion, further corroborating hydrogel's function in maintaining plasma membrane functionality.

## Discussion

In summary, intracellular assembly of hydrogel polymers was made possible through photoactivated cross-linking, presenting a unique cellular fixation strategy that seamlessly bridges the robustness of synthetic materials with the biochemical complexity of natural cells. In contrast to common fixation techniques based on chemical fixatives, intracellular hydrogelation avoids cross-linking of membrane-bound components, preserving fluid and functional plasma membrane interfaces for biological interactions. Several studies have previously examined de novo generation of globular and filamentous hydrogels in cells to mimic RNA granules[29], stimulate the phase transition of RNA/protein bodies[30], and induce cellular apoptosis[31]. However, cellular fixation and cell membrane preservation were not observed in these works. The present work differs from the aforementioned approaches in that the rapid, photoactivated assembly of covalently bonded hydrogel networks permits fast cytosolic immobilization, thereby enabling cellular fixation while obviating cellular reorganization and other cellular responses. Future development of the gelated cellular systems could further benefit from the growing arsenal of membrane manipulation strategies for intracellular hydrogel delivery[32], which may be less disruptive as compared to the freeze–thaw approach adopted in the present work. Toward biomimetic materials engineering, intracellular hydrogelation permits facile preparation of stable, cell-like constructs, offering a robust platform for device development. As cell membranes are a widely present interface with broad biological implications, the present system sees broad potential applications in biomembrane research and biomaterials engineering.

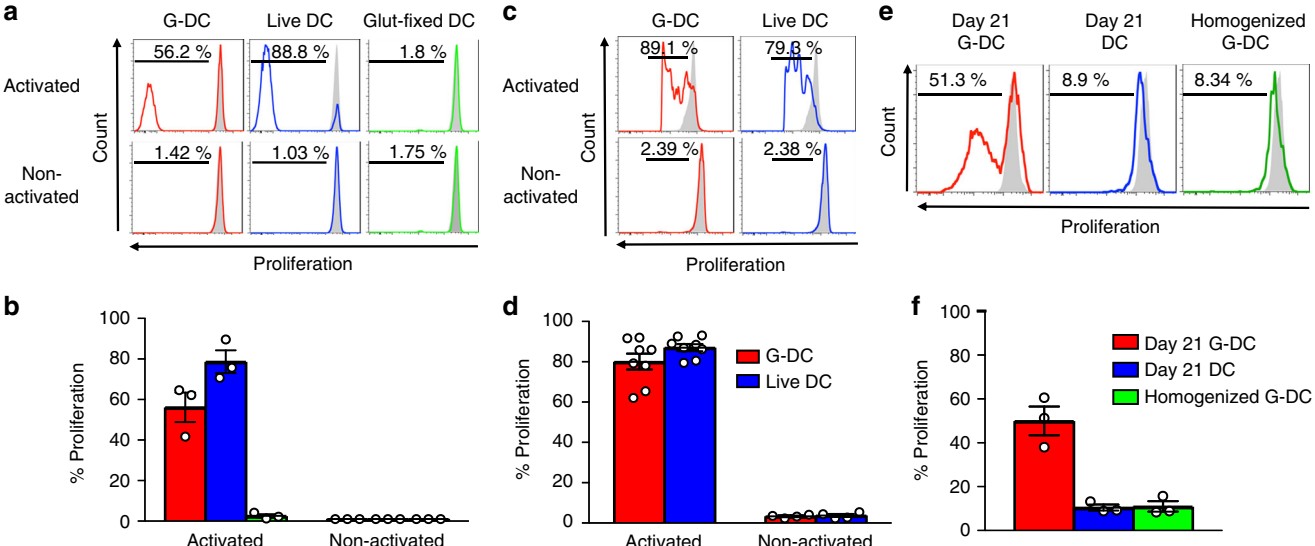

**Fig. 7** Expansion of antigen-specific T cells ex vivo and in vivo by G-DCs. **a**, **b** Antigen-specific CD8+ T-cell stimulation by G-DCs, live DCs, and glutaraldehyde-fixed DCs ex vivo. Error bars represent mean ± SEM. n = 3. **c**, **d** Antigen-specific CD8+ T-cell stimulation by G-DCs and live DCs in vivo. Error bars represent mean ± SEM. (n = 4 to 8). **e**, **f** G-DCs and non-gelated DCs were stored in PBS at 4 °C for 21 days (Day 21 G-DC and Day 21 DCs). In vivo antigen-specific CD8+ T-cell stimulation was examined with day 21 G-DC, day 21 DCs, and day 21 G-DC structurally disrupted by ultrasonication (homogenized G-DCs). T-cell expansion was monitored on day 6. CFSE-labeled T-cell groups (gray) are shown as negative controls. Error bars represent mean ± SEM, n = 3

## Methods

**Ethics statement**. All animal experiments were carried out in strict accordance with the recommendations from the Guidebook for the Care and Use of Laboratory Animals (published by The Chinese Taipei Society of Laboratory Animal Sciences). The experiment protocol was approved by the Academia Sinica Institutional Animal Care & Utilization Committee, Academia Sinica, Taipei, Taiwan.

**Cell culture**. HeLa cells, a human epithelial cell line (ATCC, CCL-2), were grown in complete media (Eagle's Minimum Essential Medium and 10% fetal bovine serum (FBS)). JAWSII cells, an immature murine dendritic cell line (ATCC, CRL-11904), were grown in complete media (alpha minimum essential medium with ribonucleosides, deoxyribonucleosides, 4 mM L-glutamine, 1 mM sodium pyruvate, 5 ng ml$^{-1}$ murine granulocyte-macrophage colony-stimulating factor, and 20% FBS).

**Intracellular gelation of suspension and adherent cells**. Gelation buffers were first prepared by mixing protease inhibitors (Pierce$^{TM}$ Protease Inhibitor Mini Tablets; ThermoFisher), 1 wt% of 2-hydroxy-4′-(2-hydroxyethoxy)−2-methylpro-piophenone (Irgacure D-2959; Sigma-Aldrich), and poly(ethylene glycol) diacrylate (PEG-DA; Mn = 700 Da; Sigma-Aldrich) ranging from 4 to 40 wt% in 10 mM phosphate buffer. For fluorescent labeling of the hydrogel network, the gelation buffers were supplemented with 0.05 wt% of fluorescein O,O'-diacrylate (Sigma-Aldrich). For cross-linking cells in suspension, adherent cells (i.e., HeLa and JAWSII cells) were detached using an enzyme-free cell dissociation buffer (ThermoFisher). Cells in suspension were pelleted at 200×g and resuspended in designated gelation buffers. For cross-linking adherent cells, cells grown on a tissue culture plate were washed with PBS and immersed in the designated gelation buffers. Immediately following the addition of gelation buffers, the cells were flash frozen in methanol precooled in a −80 °C freezer. After 10 min of freezing, the cells were thawed in a 37 °C water bath. The suspension cells were pelleted at 200×g and resuspended in PBS on a tissue culture plate, whereas adherent cells were washed with PBS twice. The tissue culture plates were then placed in an ice bath, and the cells were crosslinked with 365 nm UV wavelength for 10 min using a UV lamp (UVP UVLMS-38 EL Series) placed 2 in. above the tissue culture plate. The resulting GCs were washed twice in PBS for further experiments.

**Quantification of intracellular PEG-DA concentrations**. Quantification of intracellular PEG-DA concentrations was performed using an iodine-based quantification method[33]. Briefly, following PEG-DA infusion, 1 × 10$^6$ HeLa cells were washed and suspended in PBS to 1 mL. The collected cells were then sonicated in a bath sonicator for 1 min to release the entrapped PEG-DA, and the cellular debris was spun down via centrifugation at 3000×g for 5 min. The supernatants were collected and mixed with BaCl$_2$ and iodine solutions in an 8:2:1 ratio. Following color development for 15 min, PEG-DA concentrations in the samples were determined by measuring the light absorbance at 535 nm. A standard

curve was prepared with serially diluted PEG-DA. The measured PEG-DA content was then divided by the total volume of HeLa cells to determine the intracellular PEG-DA concentration.

**Atomic force microscopy and elastic moduli assessment**. Elastic moduli were assessed using a Zeiss axiovert microscope and analyzed using JPK NanoWizard 3 (JPK instrument, Berlin, Germany). Glass slides with different samples were attached to the tip of AFM cantilever with force constant of 0.08 N m$^{-1}$ and a resonance frequency of 20 kHz (NANOSENSORS$^{\text{™}}$). Elastic moduli of different cellular samples were quantified by contact mode. The force scanning technique was also used to generate high-resolution (64 × 64 points) topographical/elastic maps of the cells.

**Cell roundedness and fluorescence quantification**. Cell circularity was measured and calculated by ImageJ software. A built-in option for analyzing roundness is available. Briefly, after image files were imported into ImageJ software, and we chose the built-in options to enhance contrast, to analyze circularity, and to export results for cell roundness. For quantification of fluorescence intensity, fluorescent image files were processed in ZEN Imaging Software (Carl Zeiss).

**FITC-dye exclusion assay**. Totally, 1×10$^6$ gelated HeLa cells (4 wt%) were suspended in 500 μL of PBS solution containing 10 μg mL$^{-1}$ of FITC for 2 h. Totally, 50 μL of samples were then added to a confocal dish and observed by a confocal microscope.

**Hemagglutination of avian erythrocytes**. A/PuertoRico/8/34(H1N1) was propagated in 10-day-old specific-pathogen-free (SPF) chicken embryos (JD-SPF Biotech, Miaoli, Taiwan) via the allantoic route. Native virions were then derived by purifying the virus-containing allantoic fluid (AF) through 20–50% sucrose gradient solution. Avian erythrocytes were prepared from chicken whole blood upon removal of plasma and buffy coat following centrifugation at 200 × g. Gelated avian erythrocytes were prepared using 20 wt% PEG-DA. Hemagglutination study was performed by adding 10$^8$ virions to 1 mL of PBS solution containing 2% of avian erythrocytes. Presence of hemagglutination was monitored following 30 min of incubation at room temperature.

**Transmission electron microscopy**. Cellular samples were fixed using 2% glutaraldehyde in 0.1 M cacodylate buffer at pH 7.4 overnight at 4 °C. After postfixation in 1% osmium tetroxide and pre-embedding staining with 1% uranyl acetate, tissue samples were dehydrated and embedded in Agar 100. Sections measuring 80 nm were then examined using an FEI Tecnai G2 TF20 Super TWIN microscope equipped with a field emission gun.

**Cryogenic scanning electron microscopy**. For cryogenic scanning electron microscopy (cryo-SEM) imaging, an FEI Quanta 200/Quorum PP2000TR FEI, 2007 high-resolution SEM was used. Briefly, HeLa cells were seeded on Aclar embedding films for 24 h prior to PBS or gelation treatments. Before imaging, the samples were washed with PBS and suspended in RO water for freezing by liquid nitrogen. The samples were then etched under vacuum and imaged at an acceleration voltage of 3 kV by cryo-SEM.

**Examination of membrane order in GCs**. Examination of membrane lipid order in GCs was carried out according to a previously described protocol[21]. Briefly, GCs and control cells were stained in media containing 100 μg ml$^{-1}$ of Laurdan dye for 1 h. The samples were subsequently washed with PBS, and the images for membrane order analysis were acquired by confocal microscopy. For the imaging setup, the excitation wavelength was set at 405 nm, and the detection wavelengths were set at 440–460 nm for the ordered channel and 490–510 nm for the disordered channel. All images were exported in the TIFF format and saved as 32-bit grayscale image files with ImageJ. The custom-written macro provided by Owen et al.[21] was loaded into ImageJ to calculate GP values as well as to create pseudo-colored GP-intensity-merged images and intensity histograms. GP values were calculated according to the equation: GP value = $(I_{440-460\ nm} - I_{490-510\ nm})/ (I_{440-460\ nm} + I_{490-510\ nm})$, where $I$ indicate intensity of pixels.

**Fluorescence microscopy and FRAP analysis**. Cell membrane was stained by adding 10 μL of DiD dye solution (1,1′-Dioctadecyl-3,3,3′,3′-tetra-methylindodicarbocyanine; ThermoFisher Scientific) containing 5 μg mL$^{-1}$ of DiD dye and 0.5% of DMSO to 200 μL of cell suspension. HeLa cells expressing EGFP-GPI (Addgene, pCAG: GPI-GFP, #32601), CD80-GFP (Sino Biological Inc., pCMV3-mCD80-C-GFPSpark, MG50446-ACG), Transferrin Receptor (TfR) (Sino Biological Inc., pCMV3-hTfR-C-DDK (flag) tag, HG11020-CF), Lyn-GFP (Sino Biological Inc., pCMV3-hLyn-C-GFPSpark, HG10829-ACG), EGFR-GFP (Sino Biological Inc., pCMV3-mEGFR-C-GFPSpark, MG51091-ACG), and GFP-beta-actin (Sino Biological Inc., pCMV3-hbeta-actin-N-GFPSpark, HG10962-ANG) were prepared via transfection. Plasmids were transfected into cells with Lipofectamine 3000 (Invitrogen) according to the manufacturer's instruction. After 48 h, the cells were either gelated or treated with 2.5% glutaraldehyde for 10 min prior to fluorescence microscopy or FRAP analysis. Fluorescence microscopy and FRAP analysis were carried out on a Zeiss LSM780 confocal microscope (Carl Zeiss, Oberkochen, Germany) equipped with Plan-Apochromat 100×/1.4 oil objective. For FRAP analysis, adherent cells and adherent GCs were used rather than suspension cells to minimize artifacts due to random movements. An objective heater was used to maintain samples at 37 °C. Images were collected with a pinhole of 1.52 AU (1.1 μm section) for optimal signal intensity. The sample was first scanned three times with 5% of laser power to measure the fluorescence intensity before photobleaching, followed by 500 iterative laser pulses at full power to photobleach a 27 nm × 6 nm rectangular area at the plasma membrane. Fluorescence recovery was monitored every 2 s for at least 2 min at 60 frames per second until a plateau is reached. Fluorescence intensity vs. time was plotted for analyzing the fluorescence recovery. The mobile fraction was calculated based on the equation $(I_E - I_0)/(I_I - I_0) \times 100\%$, where $I_E$ is the end value of the recovered fluorescence intensity, $I_0$ is the first post-bleach fluorescence intensity, and $I_I$ is the initial (prebleach) fluorescence intensity. The halftime of recovery ($t_{1/2}$) is derived as the time from the bleach to the time point where the fluorescence recovery reaches 50% of the final recovery intensity. For the recovery rate of CD80-GFP, the rate constant $k$ was derived by converting the fluorescence recovery to a first-order elimination kinetics curve in which concentration $A_{(t)}$ is calculated as $I_I - I_{(t)}$ with $I_{(0)}$ being set as the first postbleach fluorescence intensity. The first 20 time points were used for calculating the recovery kinetics. Recovery kinetics were calculated based on the equation $\ln[A_{(t)}] = -kt + \ln[A_{(0)}]$, in which $k = -(\ln[A_{(t)}] - \ln[A_{(0)}])/t$.

**Protein tracking by TIRF microscopy**. Movements of CD80-GFP were observed by TIRF microscopy using Leica TIRF MC inverted fluorescence microscope equipped with HCX PL-APO 100× NA 1.46 Oil objective lens (Leica Microsystems, Germany). Cell samples were loaded onto a glass bottom culture plate, and the samples were exposed to a 488-nm wavelength laser. The fluorescence image was acquired using a hamamatsu EM-CCD camera (C9100-13) at a temporal resolution of 63 ms. All single-molecular experiments were performed at 37 °C. Protein movements were then analyzed using two-dimensional trajectories of CD80-GFP molecules in the plane of the basal membrane and were reconstructed by Imaris Image Analysis Software (Bitplane, Switzerland). The diffusion constants were evaluated based on a previously described method[24]. Briefly, the mean square displacement (MSD) was plotted from each trajectory against time ($t$). For each molecule, the slope of the first three time points in the MSD $t$ plot was used to calculate the diffusion coefficient, $D$, according to the equation $MSD_{t>0} = 4Dt$. Statistical analysis was performed using one-way ANOVA with GraphPad Prism. The $F$ value is 5.158, and the degrees of freedom is 5.

**Dendritic cell preparation and analysis**. For activation, JAWSII cells were seeded onto a 100 mm petri dish with 10 mL of media at a density of 10$^6$ per dish. Three

days after seeding, the cells were treated with 1 μg mL$^{-1}$ of LPS for 16 h at 37 °C and then pulsed with 10 μg ml$^{-1}$ SIINFEKL OT-I peptide for 4 h in complete media. Activated or nonactivated DCs were gelated and then resuspended in PBS with 10% FBS. FITC-conjugated hamster anti-mouse CD3 (BioLegend, #100306, clone 145-2C11, 1:100), CD80 (eBioscience, #11–0801–81, clone Ly-53, 1:100), or allophycocyanin-conjugated anti-MHC class I-SIINFEKL antibodies (eBioscience, #17-5743-80, clone25-D1.16, 1:100) were added and incubated with the cells at room temperature for 30 min in the dark. The cells were then washed twice, and the expression of surface markers was acquired by FACSCanto (BD Biosciences) and analyzed by FlowJo software (Tree Star). Gating strategies for all flow cytometric analyses are shown in Supplementary Figure 17. Statistical analysis was performed based on a two-tailed, unpaired $t$ test using GraphPad Prism.

**T-cell isolation and fluorescence labeling**. OT-I cells (CD8+ T cells specific for OVA257–264 peptide in the H2-K$^b$ context) were isolated from OT-I transgenic mice, which were a gift from Dr. Nan-Shih Liao from the Institute of Molecular Biology, Academia Sinica. After mice were sacrificed, their spleens were removed and placed into RPMI1640 complete medium with 10% FBS. In order to harvest single splenocytes, the spleens were tamped and strained with the tip of a 5 ml syringe against a sterile 40 μm nylon cell strainer (BD Biosciences Falcon, #352340). Splenocytes were incubated with BD Pharm Lyse lysing buffer (BD Biosciences, # 555899) for 3 min to remove RBCs. OT-I cells were subsequently isolated from the splenocytes using a Mouse CD8a$^+$ T Cell Isolation Kit (BD Biosciences, #19853 A). OT-I cells were stained with carboxyfluorescein diacetate succinimidyl ester (CFSE) by incubating the cells with PBS containing 5 μM of CFSE (Sigma-Aldrich, #21888) at 37 °C for 5 min. The cells were washed three times with complete medium. CFSE-labeled cells were harvested for further experimental studies.

**Examination of G-DC/T-cell interaction**. For observation of G-DC/T-cell interactions, adherent JAWSII DCs were gelated using 4 wt% PEG-DA and subsequently stained with CellTracker$^{TM}$ Deep Red dye (Molecular Probes) at 37 °C for 30 min. Stained G-DCs were washed twice using PBS and resuspended in RPMI1640 complete media supplemented with 10% FBS. CFSE-labeled OT-I cells were subsequently added to the G-DCs. The interaction between G-DCs and OT-I CD8 T cells was subsequently imaged using a confocal microscope (Zeiss LSM780 confocal microscope system, Zeiss) and analyzed using LSM Image Browser software (Zeiss). To examine CD80 clustering on G-DCs, JAWSII DCs were transfected with CD80-GFP plasmids using the TransIT-TKO transfection reagent (Mirus, #2154) following a previously described protocol protocol[34]. Briefly, plasmids were prepared using a Qiagen Plasmid Midi kit (QIAGEN, #21243). Transfection mixtures consisting of 5 mL of serum-free DMEM, 20 μg of plasmids, and 40 μL of transfection reagent were prepared and transfected into JAWSII DCs. Following 4 h of incubation, an additional 10 mL of complete medium was added to the cells. 48 h after transfection, CD80-GFP-expressing JAWSII cells were gelated with 4 wt% PEG-DA with and used for examining CD80 clustering upon incubation with antigen-specific T cells.

**T proliferation assay ex vivo**. CFSE-labeled OT-I cells were co-cultured with live DCs, G-DCs or glutaraldehyde-fixed DCs at different ratios. Co-cultured cells in 96-well v-bottomed plates were cultured at 37 °C for indicated time periods. After harvesting, cells were stained with allophycocyanin-conjugated rat anti-mouse CD8a antibodies (eBioscience, # 100712, Clone 53-6.7, 1:100) and analyzed by flow cytometry. Proliferation analysis platform in FlowJo was used to analyze cell division. For experiments involving stored G-DCs and DCs, G-DCs, and DCs were stored in PBS at 4 °C for 21 days. Homogenized G-DCs were prepared by sonicating 21-day-old G-DCs using a Fisher Scientific 150E Sonic Dismembrator at 80% power pulsed (3 s on/1 s off) for 1 min. The G-DC, DC, and glutaraldehyde-fixed DC samples were derived from the same cell source for each separate experiment.

**T proliferation assay in vivo**. CFSE-labeled splenocytes were adoptively transferred via tail vein injections to 8-week-old C57BL/6 J mice at a cell number of $3.3 \times 10^7$. Twenty-four hours after the adoptive transfer of OT-I cells, the mice were challenged with live DCs, G-DCs or Glut-fixed DCs at a cell number of $10^6$ via tail vein injections. 3 days after the DC injections, splenocytes were harvested from the mice and stained with allophycocyanin-conjugated rat anti-mouse CD8a antibodies, followed by flow cytometry and FlowJo analysis. The animal protocol was approved by the Institutional Animal Care and Use Committee (IACUC) at Academia Sinica. The G-DCs and DCs were derived from the same source of activated or nonactivated DCs.

**Reporting summary**. Further information on experimental design is available in the Nature Research Reporting Summary linked to this article.

## Data availability
All relevant data are available from the authors and/or are included within the manuscript and Supplementary Information.

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

## Acknowledgements

The authors acknowledge technical support from Common Equipment Core, Institute of Biomedical Science, Academia Sinica, for the confocal microscopy image acquisition and FRAP analysis, Yi-Ru Li for the technical support on atomic force microscopy, Kung-Hsuan Lin and Tzu-Ling Wu for the TIRF image acquisition, Yao-Kuan Huang for the technical support on transmission electron microscopy. The authors acknowledge funding support from the Academia Sinica Career Development Award (CDA-105-L06) and by the Ministry of Science and Technology, Taiwan (106-2119-M-001-010).

## Author contributions

J.C.L., C.Y.C., Y.I.C., H.W.C. and C.M.J.H. conceived the experimental designs. J.C.L., C.Y.C., Y.I.C., J.Y.C., B.Y.Y., N.N.L., Z.S.F. and W.Y.C. performed the optimization and characterization of the intracellular hydrogelation protocol. C.Y.C., Y.I.C., C.L.L., B.Y.Y. and W.Y.C. performed the membrane fluidity analysis. J.C.L. and Y.H.L. performed the immunological assays. J.C.L., C.Y.C., C.L.L. and C.M.J.H. prepared the paper. All authors have read and approved the paper.

## Additional information

**Competing interests:** The authors declare no competing interests.

