## [Peer Review File · Nature Communications]

Reviewers' comments:

Reviewer #1 (Remarks to the Author):

In this manuscript, the authors reported an effective method to fix cells while preserve the fluidity and biological function of plasma membrane. Even though intracellular hydrogelation has been studied, the research of plasma membrane after hydrogel formation was seldom reported. Thus, although the method (photo-activation) used by the authors to generate intracellular hydrogel is not novel enough and with limitation, their study on the fluidity and biological function of plasma membrane after hydrogelation are interesting. However, the present form of manuscript needs major improvement before it can accepted. The authors need to address the following points:

1. In the introduction, the authors mentioned three criteria to evaluate the intracellular hydrogel network: "(i) Hydrophilic crosslinking monomers with a low molecular weight were used to facilitate cytoplasmic permeation and minimize membrane partitioning. (ii) A crosslinking chemistry with low protein reactivity was adopted to facilitate non-disruptive cellular fixation. (iii) Extracellular crosslinking was minimized to prevent cell surface masking."

However, in this manuscript the authors poorly discussed their intracellular hydrogel network according to these three criteria. Thus, I suggest the authors to:

i. Determine and calculate the intracellular concentration of the monomer they used to prove that it can readily permeate through cell membrane.

ii. Evaluate the membrane integrity after gelation.

iii. Label cell skeletons (tubulin or actin) to check that if the intracellular hydrogel disrupt their structure. Non-disruptive cellular fixation or not? Immunofluorescent staining of cell skeletons may be difficult with intracellular hydrogel. It would be necessary to generate fluorescent cytoskeletons for such studies.

2. Since the cell morphology is stable over 30 days, I wonder how long the GCs can preserve the fluidity and biological function (e.g. antigen activity) of plasma membrane. 10 day? Or more?

3. The captions don't match in Figure 2. The caption of Figure 2C (recovery kinetic) is missing.

4. In Figure S7B, I wonder why DiD dye only stained the plasma membrane for live cells and glutaraldehyde-fixed cells, but stained the cytoplasm in GCs.

5. The authors should mention the crosslinker densities in Figure 1C.

6. Although the G-DCs seem to preserve their morphology on 21 days (figure S10C), the plasma membranes of G-DCs look different from the live cells in Figure S8A. Will the gelation damage the plasma membrane?

7. The hydrogel components were introduced into cells through membrane poration which diminish the significance of this work. This point needs to be discussed or clarified.

8. Line 99-101, the authors claim that "Close examination of the GCs by Z stacked fluorescence microscopy revealed distinctive fluorescent filaments consistent with the patterns of actin cytoskeleton" However, it looks like the filaments are outside of the cell. The authors should provide the bright field image to prove the location of these filaments.

9. For the surface markers on gelated dendritic cells and T cell proliferation assay, an important control experiments with the fixed cells should be provided.

Reviewer #2 (Remarks to the Author):

In this paper, Lin et al demonstrate what is essentially a new fixation method - the creation of an intracellular hydrogel, which they say preserves and supports the plasma membrane, allowing imaging experiments on that structure in fixed cells. Overall the method sounds interesting and I would generally support publication. However, the plasma membrane is extremely complex and I don't think the authors have sufficiently characterised the effect of their gelation method on membrane properties which may be of interest for researchers. I therefore recommend a major revision, but with the caveat that my own personal opinion is that several membrane properties might be affected and that would make the method less useful and possibly preclude publication in

NComm. The required experiments are also quite time consuming and in some cases complex.

For example, a major influencer of membrane protein diffusion is the so called picket fence model by Kusumi et al. How does their hydrogel interact with this model? I don't think their FRAP experiments (which are quite crude) can say. In theory, a dense polymer mesh below the PM would cause increased "hop diffusion" behaviour when viewed on the single-molecule level - due to a dramatic increase in the number of barriers. This should be more evident in proteins which have large intracellular domains compared to proteins which are largely in the extra cellular leaflet.

Also - how is the so called lipid raft hypothesis affected? Many experiments use environmentally sensitive dyes such as Laurdan and others to record the hydration of membrane lipids as a proxy for membrane lipid order. Do the hydrogels change this? What about the properties which are supposedly regulated by the picket fence model and lipid rafts - such as membrane protein aggregation/clustering? Is this affected? I note this is a particular feature in T cell - APC interactions.

Does the membrane remain equally porous to various substances? Are channels and pores still able to open and close within the membrane for example? What about channels that have a large intracellular domain which would interact strongly with the gel?

Are sub-resolution membrane features - e.g. ruffles also preserved?

In general, a range of proteins would need to be investigated. Kusumi et al for example investigated different fixation methods on protein diffusion in the PM and found differences in behaviour between e.g. GPI anchored proteins, transmembrane proteins and so on. Similar characterisation is needed here.

Reviewer #3 (Remarks to the Author):

Lin et al. describe a technique to achieve gelation inside cells without compromising properties and functionalities at the plasma membrane. For this, the cells are porated with flash freezing to introduce chemicals that polymerize in cells upon subsequent UV irradiation. The concept of "subcellular fixation" is interesting in contrast to conventional fixation which affects the entire cell. However, the experiments often lack control conditions (both positive and negative ones). And most importantly the study lacks direct evidence of gelation of the intracellular milieu. The authors measured elasticity of cell surface using AFM, provided EM images, and stained with a dye for polymer networks. These by themselves do not prove that there was gelation inside cells, especially when they failed to associate with control conditions. The molecular sieving property of gels should be also demonstrated in the cytosol. Another concern is no phenotype at the plasma membrane. Plasma membrane is closely associated with cytosolic components such as stress fibers and actin cortex. Intracellular gelation should impact these peri-plasma membrane components, and then plasma membrane properties, at least to some extent. The authors should assess any effect of the gelation on these peri-plasma membrane components.

Missing control conditions:

- Fig. 1B-G need controls where no chemicals were added, or chemicals were added with no UV irradiation.
- Fig. 2 needs a control with a "cytosolic" molecule that has clearly been affected by intracellular gelation.
- Fig. 3 E-G also lacks control experiments. Association with cells should be quantified for the experimental and control conditions.

Other miscellaneous points:

- Introductory sentences citing references 1-16 in the Introduction section should be more focused

to the theme of the present work. Along the same line, these references (PMID: 28041848, PMID: 29115293, DOI: 10.1002/adma.200701971) should be cited due to the close focus on intracellular gelation. Interestingly, Yang et al. (10.1002/adma.200701971) found that chemical gelation in cells leads to cell death, thus one would question the reason why the present work did not lead to cell death. This should be discussed.

- It would be helpful to show chemical structures of the PEG-DA and Irgacure D-2959, perhaps in the supplementary data.
- It would be helpful to have an explanation of the principle of fluorescein-diarylate as a gel marker.
- Typos should be corrected.

Reviewers' comments: 2018-08-02

Reviewer #1 (Remarks to the Author):

In this manuscript, the authors reported an effective method to fix cells while preserve the fluidity and biological function of plasma membrane. Even though intracellular hydrogelation has been studied, the research of plasma membrane after hydrogel formation was seldom reported. Thus, although the method (photo-activation) used by the authors to generate intracellular hydrogel is not novel enough and with limitation, their study on the fluidity and biological function of plasma membrane after hydrogelation are interesting. However, the present form of manuscript needs major improvement before it can accepted. The authors need to address the following points:

1. In the introduction, the authors mentioned three criteria to evaluate the intracellular hydrogel network: “(i) Hydrophilic crosslinking monomers with a low molecular weight were used to facilitate cytoplasmic permeation and minimize membrane partitioning. (ii) A crosslinking chemistry with low protein reactivity was adopted to facilitate non-disruptive cellular fixation. (iii) Extracellular crosslinking was minimized to prevent cell surface masking.”

However, in this manuscript the authors poorly discussed their intracellular hydrogel network according to these three criteria. Thus, I suggest the authors to:

i. Determine and calculate the intracellular concentration of the monomer they used to prove that it can readily permeate through cell membrane.

We appreciate the reviewer’s suggestions and significantly revised the manuscript to examine and discuss the three criteria of emphasis. Regarding hydrogel permeation through cell membrane, we added an additional study to quantify PEG-DA content entrapped in the cytosolic compartment of cells before and after the freeze-thaw process (**Figure 1B**). We showed that the content of hydrogel monomer in the cytosol increased significantly following the free-thaw process and matched the input monomer concentrations. The study validates that the membrane poration process allowed for rapid equilibration of hydrogel monomers inside the cells and subsequent treatments retained these monomers intracellularly. Relevant discussion has been added to the revised manuscript.

ii. Evaluate the membrane integrity after gelation.

Regarding the membrane integrity, we have added a dye-exclusion study. We used gelated cells with 4% hydrogel crosslinking, which is a composition with the highest hydrogel porosity, and showed that the gelated cells effectively excluded a FITC dye (**Figure 1F**). The study indicates that the gelated cells retain membrane integrity. Other additional studies on membrane orderness, membrane protein mobility, and membrane protein clustering also revealed retention of membrane properties on the gelated cells. We have added relevant discussion to highlight the attribute brought up by the reviewer.

iii. Label cell skeletons (tubulin or actin) to check that if the intracellular hydrogel disrupt their structure. Non-disruptive cellular fixation or not? Immunofluorescent staining of cell skeletons may be difficult with intracellular hydrogel. It would be necessary to generate fluorescent cytoskeletons for such studies.

We have followed the reviewer's suggestion and performed the recommended study using HeLa cells transfected with actin-GFP. We observed that following intracellular gelation, cells retained filamentous cytoskeletal structures, indicating that the gelation process did not disrupt the cytoskeletons (**Figure 2E and Figure S7A**). In addition, we observed that at high crosslinking percentages, cytoskeletal structures remained observable 24 h after the gelation process (**Figure S7A**), indicating high-percentage crosslinking could be applied to preserve intracellular molecular organization whereas low-percentage crosslinking permitted higher mobility to the intracellular proteins. Relevant discussion has been added to the revised manuscript.

2. Since the cell morphology is stable over 30 days, I wonder how long the GCs can preserve the fluidity and biological function (e.g. antigen activity) of plasma membrane. 10 day? Or more?

We appreciate the reviewer's comment and performed the recommended study as suggested using GCs stored at 4°C. Firstly, using EGFP-GPI-transfected HeLa cells, we showed that the membrane proteins remained mobile with negligible changes in recovery kinetics 3 days following intracellular gelation for both 4% and 40% gelated cells (**Figure S15A-D**), indicating that the protein mobility can be preserved long-term. Beyond 3 days, however, two challenges emerged: 1. EGFP fluorescence was no longer observable as the protein has a typical half-life of 22 hours and likely degraded beyond observation after 3 days. 2. Since confocal culture plates were not coated with adhesive coatings (i.e. polylysine), GCs of low hydrogel densities started to detach from confocal culture plates at day 5. However, 40 wt% GCs remained attached. We therefore performed a study with 40 wt% gelated HeLa cells transfected with transferrin receptors, which allowed us to monitor fluorescence with exogenously added, fluorescently labelled transferrin. In this study, the transferrin receptors exhibited a slight reduction in recovery kinetics but remained highly mobile 13 days after the gelated cell preparation (**Figure S15E,F**). The reduction in recovery kinetics was likely attributable to protein denaturation in aqueous solution that also accounted for the loss of GFP fluorescence. Together with the functional study that showed gelated dendritic cells remained capable of expanding antigen-specific T cells after 21 days of storage at 4°C (**Figure 7E,F**), we can reasonably conclude that the GCs can preserve mobile membrane proteins long-term. A caveat lies, however, in the protein stability upon storage. Proper storage upon freezing or lyophilization should allow functional, mobile proteins to be preserved indefinitely. Relevant discussion has been added to the manuscript.

3. The captions don't match in Figure 2. The caption of Figure 2C (recovery kinetic) is missing.

We appreciate the reviewer for bringing the error to our attention. We have made sure the captions match each figure in the revised manuscript.

4. In Figure S7B, I wonder why DiD dye only stained the plasma membrane for live cells and glutaraldehyde-fixed cells, but stained the cytoplasm in GCs.

For the reviewer's information, DiD is a highly lipophilic dye that would stain all membranous compartments of a cell upon proper solubilization in DMSO. In practice, however, DiD staining is influenced by dye penetration. As we used a minimal DMSO

content (0.5%) to minimize solvent influence on the cell membranes, DiD penetration was variable. The staining, however, would not influence the outcome or the conclusion of the study.

5. The authors should mention the crosslinker densities in Figure 1C.

We appreciate the reviewer's comment and have made sure to specify crosslinker densities in all our revised figures.

6. Although the G-DCs seem to preserve their morphology on 21 days (figure S10C), the plasma membranes of G-DCs look different from the live cells in Figure S8A. Will the gelation damage the plasma membrane?

We appreciate the reviewer's comment and would like to bring up that the differences between the original Figure S10C (**Figure S16B** in the revision) and Figure S8A (**Figure S10A** in the revision) were due to one being detached, suspension G-DCs (**Figure S16B**) and the other being adherent DCs (**Figure S10A**). For FRAP studies, gelated cells had to be prepared in their adherent forms to minimize cellular movements and drifting during examination. For T cell expansion, suspension gelated cells were prepared. Preservation of the plasma membrane was also validated in other studies, including dye exclusion study (**Figure 1F**), membrane orderness study (**Figure 3D-F**), and membrane fluidity studies (**Figure 3A-C** and **Figure 4**).

7. The hydrogel components were introduced into cells through membrane poration which diminish the significance of this work. This point needs to be discussed or clarified.

We appreciate the reviewer for bringing up such point. The present study was designed to demonstrate for the first time that intracellular hydrogelation can be an effective way to preserve fluid and functional cell membrane interface. We therefore elected to use the most straight-forward way to introduce hydrogel monomers. In future iterations, alternative approaches can be devised to introduce hydrogel monomers and crosslinkers to cellular interior. Relevant discussion has been added to the revised manuscript with reference on how emerging techniques in intracellular delivery may improve the quality of GCs.

8. Line 99-101, the authors claim that "Close examination of the GCs by Z stacked fluorescence microscopy revealed distinctive fluorescent filaments consistent with the patterns of actin cytoskeleton" However, it looks like the filaments are outside of the cell. The authors should provide the bright field image to prove the location of these filaments.

We acknowledge the reviewer's comment but regret to inform the reviewer that for the high-resolution Z-stacked image shown in the present **Figure S10D**, bright-field microscopy shows up poorly due to out-of-focus issue. For the specific image, a satisfying bright-field image was thus not possible to acquire with our imaging equipment. We've made sure to include bright-field images in other revised figures (non-Z stacked) where applicable (**Figure 1F**, **Figure 2D**, **Figure 6**). Regarding reviewer's concern of the localization of the filaments in Figure S10D, a wide range of images of cells expressing GFP-labelled cytoskeleton can be found in the public domain (i.e. PLoS ONE 8(3): e59812), and the CD80-GFP patterns are

consistent with these images. We hope that the information alleviates the reviewer's concern regarding the filaments' location.

9. For the surface markers on gelated dendritic cells and T cell proliferation assay, an important control experiments with the fixed cells should be provided.

We appreciate the reviewer's comments and have added a control experiment using glutaraldehyde-fixed dendritic cells (**Figure 5F**). The result shows that in stark contrast to gelated dendritic cells, the glutaraldehyde-fixed dendritic cells lost the ability to expand antigen-specific T cells.

Reviewer #2 (Remarks to the Author):

In this paper, Lin et al demonstrate what is essentially a new fixation method - the creation of an intracellular hydrogel, which they say preserves and supports the plasma membrane, allowing imaging experiments on that structure in fixed cells. Overall the method sounds interesting and I would generally support publication. However, the plasma membrane is extremely complex and I don't think the authors have sufficiently characterised the effect of their gelation method on membrane properties which may be of interest for researchers. I therefore recommend a major revision, but with the caveat that my own personal opinion is that several membrane properties might be affected and that would make the method less useful and possibly preclude publication in NComm. The required experiments are also quite time consuming and in some cases complex.

We appreciate the reviewer's comments and would like to express that we too share the sentiment that the plasma membrane is highly complex and difficult to preserve in its entirety. It is with this notion in mind that we believe the proposed intracellular gelation technique can add to the many prior efforts in recreating cell-like membrane interfaces, which include bottom-up membrane engineering, membrane reconstitution, and chemical crosslinking. We have taken most of the reviewer's suggestions to characterize the membrane properties following intracellular gelation and proved many membrane properties are effectively preserved. We hope that the revised manuscript may help convince the reviewer of the value of the intracellular gelation approach.

For example, a major influencer of membrane protein diffusion is the so called picket fence model by Kusumi et al. How does their hydrogel interact with this model? I don't think their FRAP experiments (which are quite crude) can say. In theory, a dense polymer mesh below the PM would cause increased "hop diffusion" behaviour when viewed on the single-molecule level - due to a dramatic increase in the number of barriers. This should be more evident in proteins which have large intracellular domains compared to proteins which are largely in the extra cellular leaflet.

We appreciate the reviewer's comment on the picket fence model but regret to inform the reviewer that our laboratory lacks the specialized equipment to offer a detailed examination and discussion on the topic. We have, however, taken the reviewer's suggestion to employ membrane proteins with variously sized intracellular domains to examine how these proteins are affected by the intracellular hydrogelation. We performed FRAP studies to examine 5

differently sized membrane proteins (EGFP-GPI, CD80-GFP, TfR, Lyn-GFP, and EGFR-GFP) in gelated cells of different hydrogel densities (**Figure 4**; **Figure S10-14**). In these studies, we observed that for hydrogel densities between 4 to 20%, no significant influence was observed on the diffusion kinetics of the proteins (**Figure 4E**). At 40% hydrogel density, however, proteins with larger intracellular domains (Lyn-GFP and EGFR-GFP) showed reduced mobility (**Figure 4F**). The finding indicates that at hydrogel densities below 20%, the hydrogel matrix is sufficiently porous and has no observable influence on membrane protein mobility as examined by FRAP. At 40% density, the additional barrier meaningfully slowed down larger proteins. While the results from the FRAP study are admittedly crude, they convincingly demonstrate the feasibility of preparing inanimate, cell-like constructs with fluid membranous cover. The ability in controlling hydrogel densities may also offer a potential approach in assessing the influence between cytoskeleton densities and membrane protein mobilities. Relevant discussion has been added to the revised manuscript.

Also - how is the so called lipid raft hypothesis affected? Many experiments use environmentally sensitive dyes such as Laurdan and others to record the hydration of membrane lipids as a proxy for membrane lipid order. Do the hydrogels change this? What about the properties which are supposedly regulated by the picket fence model and lipid rafts - such as membrane protein aggregation/clustering? Is this affected? I note this is a particular feature in T cell - APC interactions.

We have taken the reviewer's suggestion and examined the membrane lipid order on the gelated cells using Laurdan dye (**Figure 3D-F**). In the study, we showed that all gelated cells with hydrogel densities between 4 and 40% retained lipid order on their plasma membrane. In contrast, control cells with no hydrogel in the cytosol lost lipid order following the freeze-thaw treatment.

For the protein clustering study, we prepared CD80-GFP-transfected gelated dendritic cells and examined its interaction with antigen-specific T cells. Following 3 hours of incubation between the gelated DCs and the T cells, we observed distinctive fluorescent clusters at the junctions between gelated DCs and T cells (**Figure 6B**), indicating CD80 clustering. Together the results highlight retention of many cell membrane properties in the gelated cells.

Does the membrane remain equally porous to various substances? Are channels or pores still able to open and close within the membrane for example? What about channels that have a large intracellular domain which would interact strongly with the gel?

We acknowledge the reviewer's interest in the membrane permeability of the gelated cells and have added a dye-exclusion experiment to demonstrate that the GCs retain intact membrane barrier (**Figure 1F**). Regarding ion channels in the gelated cells, we appreciate the reviewer for bringing up this aspect of the plasma membrane but struggle to come up with suitable experiments and discussion for the present work. Examination of a large number of ion channels would be thematically incongruous with the manuscript's focus on the preservation of fluid and functional biomembrane interface. In addition, as most ion channels are regulated by transmembrane potentials, which are lost during the gelation process, their functions or the absence thereof may not directly reflect their biochemical functionality. Given the experimental results in the present work, we do anticipate that a large portion of membrane channels to remain biochemically functional, especially on GCs with gelation densities below 20%. We will contemplate future studies to systematically assess different ion channels in gelated cells under varying conditions.

Are sub-resolution membrane features - e.g. ruffles also preserved?

We appreciate the reviewer's comment and have added SEM studies (**Figure 2D**) and additional microscopic examinations (**Figure 2E**) of gelated cells. We observed that for adherent cells, intracellular gelation preserved membrane ruffles.

In general, a range of proteins would need to be investigated. Kusumi et al for example investigated different fixation methods on protein diffusion in the PM and found differences in behaviour between e.g. GPI anchored proteins, transmembrane proteins and so on. Similar characterisation is needed here.

We have followed the reviewer's suggestion and included representative proteins examined in the work by Kusumi et al. These proteins include GPI-anchored proteins, transmembrane proteins, and proteins known to possess large cytoplasmic domains (Lyn and EGFR) (**Figure 4D**). We performed thorough characterizations and observed that all the proteins remained equivalently mobile at gelation densities below 20%. In 40% GCs, reduction in protein mobility was observed for proteins with large cytoplasmic domains (Lyn and EGFR). In contrast to Kusumi's work, which showed that transmembrane and cytoplasmic proteins (i.e. Lyn) had significantly reduced mobility following even the mildest formaldehyde treatment, the intracellular gelation approach presented herein was able to retain the mobility of such proteins. The additional characterizations also shed light on how crosslinking densities may influence membrane protein mobility.

Reviewer #3 (Remarks to the Author):

Lin et al. describe a technique to achieve gelation inside cells without compromising properties and functionalities at the plasma membrane. For this, the cells are porated with flash freezing to introduce chemicals that polymerize in cells upon subsequent UV irradiation. The concept of "subcellular fixation" is interesting in contrast to conventional fixation which affects the entire cell. However, the experiments often lack control conditions (both positive and negative ones). And most importantly the study lacks direct evidence of gelation of the intracellular milieu. The authors measured elasticity of cell surface using AFM, provided EM images, and stained with a dye for polymer networks. These by themselves do not prove that there was gelation inside cells, especially when they failed to associate with control conditions. The molecular sieving property of gels should be also demonstrated in the cytosol.

We appreciate the reviewer's comment and have added negative controls to most experimental results. Regarding the lack of direct evidence of cytosolic hydrogelation mentioned by the reviewer, we believe some of our data previously placed in the supplement may have been overlooked. In particular, direct evidence of cytosolic hydrogel was shown with gelated cells solubilized with SDS, which yielded non-dissolvable hydrogel matrices. We have moved the data to the main figures (**Figure 2A**) to better illustrate the finding. A negative control is not included for the particular figure as normal cells dissolve completely in SDS. Other supplementary figures (**Figure S4**) as well as additional studies with SEM (**Figure 2D**) and FRAP (**Figure 4**) also provide further evidence of hydrogel support in the cytosol. We believe the data in the revised manuscript unequivocally demonstrates successful intracellular hydrogelation.

Another concern is no phenotype at the plasma membrane. Plasma membrane is closely associated with cytosolic components such as stress fibers and actin

cortex. Intracellular gelation should impact these peri-plasma membrane components, and then plasma membrane properties, at least to some extent. The authors should assess any effect of the gelation on these peri-plasma membrane components.

We appreciate the reviewer's suggestion and have added SEM examinations (**Figure 2D**; **Figure S7**) and fluorescence microscopy images of actin-GFP transfected cells (**Figure 2E**; **Figure S7A**). In these studies, we observed that gelated cells retained their ruffled membrane features as well filamentous cytoskeleton. In addition, we observed that for gelated cells with high hydrogel densities, these features can be preserved over a prolonged period of time. These observations indicate that the gelation process, which is designed to take place rapidly, can retain peri-plasma membrane components.

Missing control conditions:

- Fig. 1B-G need controls where no chemicals were added, or chemicals were added with no UV irradiation.

We have taken the reviewer's suggestion and added negative controls to applicable images (**Figure 1C,D**). In other figures, the suggested controls yielded dirty or null images that do not help with the presentation of the work. We have thus expanded **Figure S4**, in which we used FITC-diacrylate infused gelated cells to assess a variety of control conditions. The figure shows that following UV-activated hydrogelation, gelated cells possess crosslinked fluorescent hydrogel cores that are resistant to SDS dissolution. Without UV crosslinking, however, the cells were readily dissolved by SDS despite hydrogel monomer infusion. These results show that the hydrogel infusion and UV-activated crosslinking are both critical for the intracellular hydrogelation process.

- Fig. 2 needs a control with a "cytosolic" molecule that has clearly been affected by intracellular gelation.

We appreciate the reviewer's suggestion and the influence of hydrogel on "cytosolic" molecule was demonstrated in two sets of additional experiments in the revised manuscript. Firstly, we vastly expanded the number of membrane proteins assessed in Figure 4 to include proteins possessing a range of differently sized cytosolic domains (**Figure 4D**). In the study, we showed that at 40% of hydrogel crosslinking, proteins with large cytosolic domains (Lyn-GFP and EGFR-GFP) showed significantly reduced mobility. In addition, in examining peri-plasma components using actin-GFP, we also observed that high hydrogel densities (20% and 40%) retained cytoskeletal features for at least 24 h. In low density gelated cells (4% and 10%), filamentous patterns became less prominent over the same time span, indicating that the actin proteins were more easily depolymerized and dissipated. These results illustrate that the intracellular hydrogel affect the movement of cytosolic molecules.

- Fig. 3 E-G also lacks control experiments. Association with cells should be quantified for the experimental and control conditions.

We have added negative controls to the indicated experiments (**Figure 5B-F** in the revised manuscript) using glutaraldehyde-fixed dendritic cells. The results show that glutaraldehyde fixation deprived DCs of functional surface molecules and T-cell expanding capacity.

Regarding quantification of cell association, we struggle to come up with a fair quantitative assessment as T cell engagement on the gelled DCs was a dynamic rather than a static process. In supplementary video 8, we show that multiple T cells dynamically engaged with gelled DCs, with each engagement lasting approximately 2-3 h. In addition, we observed CD80 clusters on gelled DCs adjacent to antigen-specific T cells (**Figure 6B**). With glutaraldehyde-fixed DCs, no active engagement was observed (**Figure 6A**). As the number of associated cells at any given time is highly dependent on the incubation condition, we feel that providing such number would be misleading. Given that quantitative values are shown extensively in T cell expansion studies in the present work, we elect to show representative figures and videos to highlight the dynamic, lifelike interactions between the T cells and gelled DCs. We also included relevant discussion to emphasize the non-static nature of the interaction.

Other miscellaneous points:

- Introductory sentences citing references 1-16 in the Introduction section should be more focused to the theme of the present work. Along the same line, these references (PMID: 28041848, PMID: 29115293, DOI: 10.1002/adma.200701971) should be cited due to the close focus on intracellular gelation. Interestingly, Yang et al. (10.1002/adma.200701971) found that chemical gelation in cells leads to cell death, thus one would question the reason why the present work did not lead to cell death. This should be discussed.

We appreciate the reviewer's suggestion and have revised the introduction accordingly. With regard to the other references brought up by the reviewer, we would like to first clarify that the gelation process in the present work did lead to cell death as gelled cells are by all definition dead cells. In comparison to these other works with intracellular hydrogels, the distinguishing aspect of the present work is the hydrogel conformation and the rapid gelation kinetics that allow for fast cytosolic immobilization and stabilization of cellular morphology. The aforementioned gels adopt either globular (PMID: 28041848 and 29115293) or fibrous forms (10.1002/adma.200701971), which would be inadequate in providing the needed internal mechanical support for stabilization. In addition, these gels have comparatively slow gelation kinetics, therefore allowing cells to undergo programmed cell death under stress. These issues are overcome with the UV-activated, covalently bonded PEG hydrogels. Relevant discussion has been added to the manuscript.

- It would be helpful to show chemical structures of the PEG-DA and Irgacure D-2959, perhaps in the supplementary data.

We have taken the reviewer's suggestion and included the chemical structures of PEG-DA and I-2959 in Figure S1.

- It would be helpful to have an explanation of the principle of fluorescein-diarylate as a gel marker.

We have also included a graphical illustration to explain the use of fluorescein-diarylate as a gel marker.

- Typos should be corrected.

We have proof-read the manuscript and made sure typos have been corrected.

REVIEWERS' COMMENTS:

Reviewer #1 (Remarks to the Author):

The authors have adequately addressed our comments. Therefore, we suggest it to be accepted without revision.

Reviewer #2 (Remarks to the Author):

The authors have conducted a number of new experiments to investigate the effect of the gelation method on membrane properties. They have addressed 90% of my suggestions for what would be required and most membrane properties seem to be largely unaffected by the process.

Overall, I think the method is potentially interesting for the community and the authors have done enough to convince me that the samples remain viable enough for interesting studies to be conducted on them. I would therefore recommend publication.

Reviewer #3 (Remarks to the Author):

The authors addressed many points in an appropriate manner. However, there are still a few reservations.

1. Hydrogels are by definition a polymer network retaining water with characteristic mechanical and biophysical properties. The characterization assays the authors performed do not address mechanical property by for example using a rheometer. Unfortunately, the presented data so far remain as circumstantial evidence.
2. References 29-31 are introduced by mentioning "Several studies have previously introduced globular and filamentous hydrogels into cells", which is a wrong statement; these works generated gels de novo.
3. Based on Fig. 2E, the authors now claim that both control and gelated cells indicate membrane ruffles. However, the presence of ruffles in the images is not readily evident. Please highlight such structures, if any, and QUANTIFY the metric.
4. In general, quantification is missing for most of the image-based data (Fig. 1D-F and Fig. 2, to name a few). Unbiased, quantitative image analysis is a crucial component of life science, and thus has to take place before its publication.

REVIEWERS' COMMENTS:

Reviewer #1 (Remarks to the Author):

The authors have adequately addressed our comments. Therefore, we suggest it to be accepted without revision.

We appreciate all the helpful comments by the reviewer.

Reviewer #2 (Remarks to the Author):

The authors have conducted a number of new experiments to investigate the effect of the gelation method on membrane properties. They have addressed 90% of my suggestions for what would be required and most membrane properties seem to be largely unaffected by the process.

Overall, I think the method is potentially interesting for the community and the authors have done enough to convince me that the samples remain viable enough for interesting studies to be conducted on them. I would therefore recommend publication.

We appreciate all the helpful comments by the reviewer.

Reviewer #3 (Remarks to the Author):

The authors addressed many points in an appropriate manner. However, there are still a few reservations.

1. Hydrogels are by definition a polymer network retaining water with characteristic mechanical and biophysical properties. The characterization assays the authors performed do not address mechanical property by for example using a rheometer. Unfortunately, the presented data so far remain as circumstantial evidence.

Response:

We appreciate the reviewer's insightful comment on hydrogel characterization and agree that the present work places more emphasis on the aggregate property of the gelled cells rather than the hydrogel component. We will keep the comments in mind and be sure to more closely examine the hydrogel in future studies.

2. References 29-31 are introduced by mentioning "Several studies have previously introduced globular and filamentous hydrogels into cells", which is a wrong statement; these works generated gels de novo.

We apologize for the improper phrasing of the statement. The statement has been revised in the discussion section of the manuscript to properly describe the works.

3. Based on Fig. 2E, the authors now claim that both control and gelated cells indicate membrane ruffles. However, the presence of ruffles in the images is not readily evident. Please highlight such structures, if any, and QUANTIFY the metric.

We appreciate the reviewer's comment. To better highlight the ruffled structures, zoomed-in images are included in the updated Fig. 2E. We believe the revised figure evidently highlights the preserved ruffles in the gelated cells.

4. In general, quantification is missing for most of the image-based data (Fig. 1D-F and Fig. 2, to name a few). Unbiased, quantitative image analysis is a crucial component of life science, and thus has to take place before its publication.

We appreciate the reviewer's comment regarding the need for quantitative image analysis. To enhance the scientific rigor as suggested by the reviewer, we have included quantitative image analysis to Fig. 1D-F and Fig. 6B in the revised manuscript. The analysis results are shown in Supplementary Figure 3, 4, and 16. The Methods section has been updated to include the quantitation methods.